# Advanced Microscopy for Liver and Gut Ultrastructural Pathology in Patients with MVID and PFIC Caused by MYO5B Mutations

**DOI:** 10.3390/jcm10091901

**Published:** 2021-04-28

**Authors:** Michael W. Hess, Iris M. Krainer, Przemyslaw A. Filipek, Barbara Witting, Karin Gutleben, Ilja Vietor, Heinz Zoller, Denise Aldrian, Ekkehard Sturm, James R. Goldenring, Andreas R. Janecke, Thomas Müller, Lukas A. Huber, Georg F. Vogel

**Affiliations:** 1Institute of Histology and Embryology, Medical University of Innsbruck, A-6020 Innsbruck, Austria; michael.hess@i-med.ac.at (M.W.H.); barbara.witting@i-med.ac.at (B.W.); karin.gutleben@i-med.ac.at (K.G.); 2Austrian Drug Screening Institute, ADSI, A-6020 Innsbruck, Austria; iris.krainer@i-med.ac.at (I.M.K.); przemyslaw.filipek@adsi.at (P.A.F.); 3Institute of Cell Biology, Medical University of Innsbruck, A-6020 Innsbruck, Austria; ilja.vietor@i-med.ac.at (I.V.); lukas.a.huber@i-med.ac.at (L.A.H.); 4Department of Internal Medicine, Medical University of Innsbruck, A-6020 Innsbruck, Austria; heinz.zoller@tirol-kliniken.at; 5Department of Paediatrics I, Medical University of Innsbruck, A-6020 Innsbruck, Austria; denise.aldrian@i-med.ac.at (D.A.); andreas.janecke@i-med.ac.at (A.R.J.); thomas.mueller@tirol-kliniken.at (T.M.); 6Department of Paediatric Gastroenterology and Hepatology, University Hospital for Children and Adolescents, University of Tübingen, D-72076 Tübingen, Germany; ekkehard.sturm@med-uni.tuebingen.de; 7Department of Surgery and the Epithelial Biology Center, Vanderbilt University Medical Center, Nashville, TN 37232-8240, USA; jim.goldenring@vanderbilt.edu

**Keywords:** Rab11a, syntaxin3, myosinVb, NHE3, organoid, recycling endosome, progressive familial intrahepatic cholestasis, congenital diarrheal disorder, high-pressure freezing, immuno-electron microscopy

## Abstract

Mutations in the actin motor protein myosinVb (myo5b) cause aberrant apical cargo transport and the congenital enteropathy microvillus inclusion disease (MVID). Recently, missense mutations in myo5b were also associated with progressive familial intrahepatic cholestasis (MYO5B-PFIC). Here, we thoroughly characterized the ultrastructural and immuno-cytochemical phenotype of hepatocytes and duodenal enterocytes from a unique case of an adult MYO5B-PFIC patient who showed constant hepatopathy but only periodic enteric symptoms. Selected data from two other patients supported the findings. Advanced methods such as cryo-fixation, freeze-substitution, immuno-gold labeling, electron tomography and immuno-fluorescence microscopy complemented the standard procedures. Liver biopsies showed mislocalization of Rab11 and bile canalicular membrane proteins. Rab11-positive vesicles clustered around bile canaliculi and resembled subapical clusters of aberrant recycling endosomes in enterocytes from MVID patients. The adult patient studied in detail showed a severe, MVID-specific enterocyte phenotype, despite only a mild clinical intestinal presentation. This included mislocalization of numerous proteins essential for apical cargo transport and morphological alterations. We characterized the heterogeneous population of large catabolic organelles regarding their complex ultrastructure and differential distribution of autophagic and lysosomal marker proteins. Finally, we generated duodenal organoids/enteroids from biopsies that recapitulated all MVID hallmarks, demonstrating the potential of this disease model for personalized medicine.

## 1. Introduction

### 1.1. Methodological Rationale

High-resolution imaging and preparation techniques for cell and tissue samples have considerably improved during the past decades. Rather than for histopathology, those methods are mainly applied to research—for obvious reasons. Biopsies from patients must be quickly processed and analyzed for diagnosis and treatment by using standard methods and instrumentation [1,2]. Research focused on the pathophysiology of the diseases is less committed in that respect. Moreover, research laboratories frequently benefit of amenable disease models and more complex instrumentation, as well as a wider range of analytical techniques. Animal or cell models and organoid cultures serve as valuable complement, overcoming the always limited availability of fresh patient samples [3,4,5,6,7,8,9,10,11,12,13,14,15,16]. Regarding subcellular morphology, cryo-based methods for sample preparation merit special attention [17,18,19]. Antibody- and nanobody-based labeling techniques substantially contribute to the precise localization of disease-relevant molecules in the structural context of cells and tissues [10,20,21,22,23,24,25,26,27].

### 1.2. Disease Investigated

Microvillus inclusion disease (MVID) is a very apt example of a congenital disease with complex and multifaceted subcellular phenotype [10,28,29,30,31]. Together with clinical parameters the histopathology and obligatory ultrastructural analysis by electron microscopy (EM) were the gold standard for a definite diagnosis [29,32,33] until the identification of disease-causing mutations of myosinVb (MYO5B), syntaxin3 (STX3) and syntaxin-binding protein2 (STXBP2) [4,11,34]. MVID is an autosomal inherited intestinal disorder of which two clinical forms have been described. An early-onset form develops within hours or days of birth, and a late-onset form, occurring in the first months of life. The typical intractable watery diarrhea leads to life-threatening dehydration and metabolic acidosis [29,32]. As for a severe diarrhea, the only treatment available is either life-long parenteral nutrition or small bowel transplantation [32]. Small and large intestine of MVID patients display severe villus atrophy. At the subcellular level the patients’ enterocytes show the following features: (i) variable loss of brush-border microvilli, (ii) subapical accumulation of different kinds of Periodic Acid Schiff (PAS)-positive vesicles and tubules (historically termed “secretory granules” [10,28,35]), and (iii) intracellular lumina lined by ectopic microvilli, termed microvillus inclusions [29,36]. Loss of function or missense mutations in MYO5B, STX3 or STXBP2 disrupt selective apical cargo transport in enterocytes. Myo5b interacts with the small GTPases Rab8a and Rab11a in order to transport, dock and fuse cargo vesicles to the apical plasma membrane. There, interaction with the exocyst subunit sec15 allows tethering of the vesicle. Subsequent vesicle fusion with the plasma membrane is mediated via a soluble N-ethylmaleimide-sensitive-factor attachment receptor (SNARE) complex formed by vamp8, synaptotagmin-like protein 4a, munc18-2 and syntaxin3 [5,10,11].

Over the last years evidence was increasing that mutations in MYO5B do not only affect the bowel but epithelial organs in general [37]. A particular focus lies on hepatocyte function and apical secretion into the bile canaliculus. Initially, elevated liver enzymes in MVID patients were often interpreted as secondary effect of intestinal failure and long-term parenteral nutrition. Yet, this clinical presentation was lately recognized as distinct variant of progressive familial intrahepatic cholestasis type 6 (PFIC6) [38]. It was shown that liver morphology in MVID patients with cholestasis resembled that of PFIC [38]. Another study then reported 10 patients with PFIC and MYO5B mutations, but without the typical intestinal symptoms associated with MVID [39]. Therefore, MYO5B mutations can result in MVID, PFIC, as well as MVID with PFIC. PFIC is characterized by neonatal cholestasis, elevated liver enzymes, normal or elevated gamma-glutamyl transpeptidase (GGT) levels and malabsorption related to cholestasis. So far, six genes have been associated with PFIC which are involved in bile canalicular membrane homeostasis, bile salt transmembrane transporters or bile salt metabolism [40,41]. Eventually biliary diversion or liver transplantation is the only curative treatment with several disease modifying compounds still in medical trials. PFIC6 is marked by low gamma-glutamyl transpeptidase, elevated liver enzymes and cholestasis [38,39,42,43]. Mislocalization of bile salt export pump (BSEP] or multidrug resistance p-glycoprotein 3 were noted in some PFIC6 patients [38,39,42]. In a recent detailed study on myo5b missense mutations and bile canalicular homeostasis in the human hepatocyte HepG2 cell line, multidrug resistance-associated protein 2 (MRP2) was found aberrantly targeted [41]. Thus, the emerging field on MYO5B associated PFIC (MYO5B-PFIC) has focused on liver function and morphology of PFIC6 patients. Concerning MVID, typical intestinal symptoms were not present in several MYO5B-PFIC patients [39,42,44]. A detailed subcellular and ultrastructural characterization of the intestine in MYO5B-PFIC patients is lacking as yet.

### 1.3. Aim of This Study

Based on our long-standing interest and experience regarding MVID, our multidisciplinary research team has established a wide range of different sample preparation methods and procedures for advanced microscopy. The specific added value for ultrastructural pathology and more generally for disease-relevant immuno-histochemistry at EM-level is highlighted here by a careful study on the corresponding disease characteristics of MVID and PFIC. Thorough investigations were performed on samples from a unique disease case of a patient with MYO5B mutations, complemented by selected data from two other relevant cases (all recently described: [44]). The case documented in detail here is remarkable because the adult patient presents only episodically with symptoms of the digestive system. Biopsies were obtained from this patient, which were also used to generate duodenal organoid (enteroid) cultures. Interestingly, we detected rather complex subcellular duodenal and liver phenotypes in this patient, despite unusually mild MVID-manifestation. The congruent observations from biopsies and enteroids were discussed for their cell biological, pathophysiological and clinical relevance. Furthermore, we suggest a few, simple methodological improvements for the preparation of valuable, sometimes unique biopsy material.

## 2. Materials and Methods

### 2.1. Biopsy Sampling

Duodenal biopsies were taken with a standard endoscopic biopsy forceps. Liver biopsies were taken with an automatic biopsy instrument (16 gauge needles, 19 mm notch length) or as wedge biopsies during biliary diversion procedure.

### 2.2. Enteroid Culture

#### 2.2.1. Reagents

Advanced DMEM/F-12 (Gibco-Thermo Fisher, Waltham, MA, USA, #12634028); Glutamax 100× (GIBCO, Waltham, MA, USA, #35050038); HEPES (1 M) (GIBCO, #15630080); Pen/Strep 100× (GIBCO, #15070063); WCM (Wnt3a Conditioned Medium) (homemade); Recombinant Human R-Spondin-1 (Preprotech, Rocky Hill, NJ, USA, #120-38); Recombinant Murine Noggin (Preprotech, #250-38); Nicotinamide (Sigma-Aldrich, St. Louis, MO, USA, #N0636-100G); N-Acetylcyteine (Sigma-Aldrich, #A9165-25G); B-27 (Thermo Fisher, Waltham, MA, USA, #17504044); m-EGF (REPROTECH, #315-09); Primocin–500 mg (InvivoGen, San Diego, CA, USA, # ant-pm-1); Y-27632 dihydrochloride (Abcam, Cambridge, UK, #ab120129); A83-01 (R&D Systems Europe, #2939); SB 202190 (Sigma-Aldrich, #S7067-5MG); DPBS, 1×, no calcium, no magnesium (Gibco, #14190-094); 0.25% Trypsin-EDTA (Sigma-Aldrich, #T4049-100ML); EDTA (Sigma-Aldrich, #431788-25G); Matrigel (Corning, Corning, NY, USA, #356231); DMSO Hybri-Max (Sigma-Aldrich, #D2650); DTT (Sigma-Aldrich, #10197777001); Fetal Bovine Serum (FBS; BioWest, Riverside, MO, USA, #S1810-500).

Washing Medium: Advanced DMEM/F12, 10 mM (2.38 mg/mL) HEPES, 1× Glutamax, Pen/strep.

Seeding Medium: Advanced DMEM/F12, 10 mM (2.38 mg/mL) HEPES, 1× Glutamax, Pen/strep, 50% WCM, 1 µg/mL Recombinant Human R-Spondin-1, 100 ng/mL Recombinant Murine Noggin, 10 mM (1.22 mg/mL) Nicotinamide, 1.25 mM (0.2 mg/mL) N-Acetylcysteine, 1× B27, 50 ng/mL mEGF, 10 µM (2.47 µg/mL) Y-27632, 500 nM (0.2 µg/mL) A83-01, 10 µM (3.31 µg/mL) SB202190, 100 µg/mL Primocin.

Expansion Medium: Advanced DMEM/F12, 10 mM (2.38 mg/mL) HEPES, 1× Glutamax, Pen/strep, 50% WCM, 1 µg/mL Recombinant Human R-Spondin-1, 100 ng/mL Recombinant Murine Noggin, 10 mM (1.22 mg/mL) Nicotinamide, 1.25 mM (0.2 mg/mL) N-Acetylcysteine, 1× B27, 50 ng/mL mEGF, 500 nM (0.2 µg/mL) A83-01, 10 µM (3.31 µg/mL) SB202190, 100 µg/mL Primocin.

Differentiation Medium: Advanced DMEM/F12, 10 mM (2.38 mg/mL) HEPES, 1× Glutamax, Pen/strep, 1 µg/mL Recombinant Human R-Spondin-1, 100 ng/mL Recombinant Murine Noggin, 1.25 mM (0.2 mg/mL) N-Acetylcysteine, 1× B27, 50 ng/mL mEGF, 500 nM (0.2 µg/mL) A83-01, 100 µg/mL Primocin.

#### 2.2.2. Generation and Culture of Patient Enteroids

Duodenal biopsies from patient #1 and a patient with an unrelated disorder serving as control (i.e., microdeletion syndrome 11p15.5) were washed several times with ice-cold 1× DPBS until the supernatant was cleared. Biopsy fragments were incubated in chelation buffer (1× DPBS + Pen/Strep + 10 mM (2.92 mg/mL) EDTA) for 45 min at +4 °C on a roller. Subsequently, chelation buffer was replaced with 2 mL of 1× DPBS + Pen/Strep + 40 mg/mL DTT. Tissue fragments were passed several times through a pipette of which the P1000 tip was truncated. The isolated crypts were transferred to 15 mL tubes containing 2 mL of FBS. The isolation step was repeated 6 times, each time with fresh 1× DPBS + Pen/Strep + 40 mg/mL DTT buffer. The presence of crypts was confirmed by inspection through an inverted microscope. Crypts were pelleted by centrifugation at 170× *g* for 3 min at +4 °C. They were re-suspended in washing medium plus 66% Matrigel and seeded as 30 µL drops into pre-warmed 24 well tissue culture plates (#662160, Greiner bio-one, Kremsmünster, Austria). Seeding medium was added 15 min later, after the Matrigel drops had hardened. After 5 days the enteroids were splitted as single cells and cultured in expansion medium. For differentiation, the expansion medium was replaced by differentiation medium, in which enteroids were then cultured for 5 days. During enteroid culture, the medium was replaced every 2–3 days. In order to collect enteroids for EM-preparation differentiation medium was first replaced with cold washing medium in which they were subsequently transferred to 15 mL tubes with a truncated P1000 pipette tip. Enteroids were allowed to sediment in the tubes for 5 min on ice. The supernatant was replaced with fresh cold washing medium and the tube was inverted several times to re-suspend enteroids. Washing was repeated 3 times. Finally, washing medium was replaced with cold 1× DPBS + 10% FBS and the enteroid suspension was transported on ice to the EM-laboratory for further processing.

### 2.3. Fluorescence Microscopy (IF)

Duodenal and liver biopsies were embedded in Tissue-Tek O.C.T. Compound (Sakura^®^ Finetek, VWR) and frozen at −80 °C as essentially described previously [7]. 10 µm-cryostat sections were fixed for 20 min in 4% buffered formaldehyde solution at room temperature (RT) and labeled with antibodies listed as follows and/or Alexa Fluor 568-conjugated phalloidin (for actin staining; 1:500, #A12380, Thermo Fisher Scientific, Waltham, USA), plus Hoechst 3342 (1:10000, Thermo Fisher Scientific). Primary antibodies were: Rab11 (1:100, #71-5300, Life Technologies), Syntaxin3 (1:100, #ab133750, Abcam), NHE3 (1:100, #HPA036669, Sigma), DPP4 (1:100, #HBB3/775/42, Developmental Studies Hybridoma Bank: DSHB), Myo5b (1:10, [3]), BSEP (1:100, #sc-74500, Santa Cruz), MRP2 (1:100, #sc-59608, Santa Cruz). Samples mounted in Mowiol 4-88 (#81381, Sigma) were analyzed with an Axio Imager M1 epifluorescence microscope (Carl Zeiss; Oberkochen, Germany) equipped with a charge-coupled device camera (SPOT Xplorer; Visitron Systems, Puchheim, Germany) and recorded with VisiView 2.0.3 (Visitron Systems). Objective lens used was a 10× air objective with a numerical aperture of 0.3 (Carl Zeiss, Oberkochen, Germany). Single confocal planes or stacks were recorded with a Leica SP5 confocal fluorescence microscope (Leica Microsystems, Wetzlar, Germany) using an oil immersion 63× lens with a numerical aperture of 1.4 (Leica Microsystems, Wetzlar, Germany). Recording software used was Leica LASAF 2.7.3 (Leica Microsystems). Images were deconvolved with Huygens Professional Deconvolution and AnalysisSoftware (Scientific Volume Imaging, Hilversum, The Netherlands), exported using ImageJ, version 1.49a software (National Institutes of Health, Bethesda, MD, USA) and adjusted for brightness, contrast and pixel size.

### 2.4. Ultrastructural Analyses (EM Approaches)

#### 2.4.1. Reagents

Chemicals were purchased from EMSdiasum (Hatfield, PA, USA): 32% (*v*/*v*) paraformaldehyde (formaldehyde) aqueous solution (# 15714-S); glutaraldehyde (#16210), osmium tetroxide (OsO_4_; #19100), epoxy resins (#14120 Embed 812, an Epon substitute), UranyLess™ staining solution (#22409), uranyl acetate (UA; #22400; note: uranyl compounds are radioactive, thus, subject to specific safety regulations as to purchase, handling, disposal); from Sigma-Aldrich: periodic acid (PA; #10450-60-9), thiosemicarbazide (TCH; #79-19-6), silver proteinate (SP; #9008-42-8), lead citrate (#15326), methyl cellulose (#M7027), paraformaldehyde powder (#158127), phosphate buffer solution (0.1 M; #5244); or from other sources (i.e., gelatin, sucrose both from food store), HQ-Silver^®^ (#2012, from Nanoprobes, Yaphank, NY, USA).

#### 2.4.2. Specimen Preparation

Techniques employed comprised chemical fixation and/or cryo-fixation followed by epoxy resin embedding, as well as cryo-ultramicrotomy for Tokuyasu-immuno-EM [20]. Their virtues and limitations are explained in detail in Appendix A, page 9–10 (protocol I–IV).

(a)Chemical fixation

We generally prefer and strongly suggest initial fixation with buffered formaldehyde solutions (FA) only. FA-fixed specimens are universally suitable for many kinds of microscopic analysis (morphology, immuno-labelling, classical cytochemistry, etc.). In samples fixed by glutaraldehyde solutions (GA), however, the reactivity to antibodies and certain other cytochemical reagents is sometimes masked or completely destroyed [45]. Thus, biopsies were immersed into FA-solution (4% (*w*/*v*), in 0.1 M sodium phosphate buffer (13.8 mg/mL NaH_2_PO_4_, 14.2 mg/mL Na_2_HPO_4_; pH 7.2; prepared from paraformaldehyde powder or, conveniently, from aqueous (32% (*v*/*v*) stock)) and fixed at RT for 24 h up to 3 days, as it is advisable for pure FA [46]. To fix enteroids suspended in 1× DPBS + 10% FBS the suspension was mixed with an equal volume of 8% (*w*/*v*) buffered FA at RT and gently shaken. Fixed biopsies and enteroids were optionally stored in 1–4% (*w*/*v*) FA at +4 °C.

(b)Standard processing for resin embedding

FA-fixed samples were post-fixed for ultrastructural investigation with buffered glutaraldehyde solution (GA; 2.5% (*v*/*v*) in 0.1 M sodium phosphate buffer, pH 7.2) for 1 h at RT. According to our long-term experience with any kind of mammalian tissue this sequence yields well preserved samples, fully equal to directly GA-fixed ones [47]. The advantage of this seemingly laborious, complicated protocol is, in our opinion, obvious—regarding the frequently stressed conditions in clinical daily life. Initial fixation with FA leaves multiple opportunities open that it is not necessary to split valuable, minute samples in advance for different purposes. Since FA-fixed samples can be stored well at +4 °C without losing quality, sometimes unique material is available for repeated or refined analyses. This is especially recommended for valuable samples from patients with orphan diseases, as patients frequently move to other (sometimes remote) locations or decease. The GA-post-fixed samples were rinsed with buffer, additionally post-fixed and en-bloc stained with unbuffered aqueous OsO_4_ solution (0.5% (*w*/*v*), 1–3 h, at RT), optionally followed by aqueous uranyl acetate solution (0.2% (*w*/*v*), overnight at +4 °C; with rinses in between and thereafter). Finally, samples were dehydrated through an ascending ethanol series, followed by 3 rinses with absolute acetone and embedded in Embed 812 epoxy resin (without using carcinogenic propylene oxide as intermediate).

(c)Cryo-fixation through high-pressure freezing (HPF: [17]) and freeze-substitution (FS: [18])

Dense suspensions of live enteroids were transferred into the cup-shaped HPF-carriers (well size: Ø 2 mm, depth: 0.1 mm, from Engineering Office M. Wohlwend GmbH, Sennwald, Switzerland) by using a pipette with truncated tip. HPF was performed with a HPM 010 apparatus (BAL-TEC, Balzers, Liechtenstein, Ref. [19]) and the frozen samples were immediately transferred into cryo-vials filled with frozen FS media (Nunc™ 1.8 mL Externally-Threaded Universal Tubes, from Thermo Fisher Scientific). FS media was acetone containing 0.5–1% (*w*/*v*) OsO_4_, 0.08% (*w*/*v*) uranyl acetate and 4.5% (*v*/*v*) water [48]. FS was carried out at about −80 to −90 °C for about 12–36 h [19] in a FS device (LEICA AFS2 from LEICA-Microsystems, Vienna, Austria; or cheaper: in a dry ice-acetone bath), followed by sample warming to RT, rinsing in acetone and infiltration with Embed 812 epoxy resin.

(d)Ultramicrotomy, staining and PAS cytochemistry of resin blocks

Sections (100 nm for transmission EM, 300 nm for EM-tomography) were cut from the heat-cured resin blocks by using an ultramicrotome equipped with a diamond knife (LEICA Ultracut S, plus DIATOME™ ultra diamond knife, from LEICA and Diatome Ltd., Nidau, Switzerland, respectively). Sections were mounted on mesh-grids or formvar-coated slot-grids and optionally post-stained with heavy metals (leads salts, or UranyLess™ staining solution) for contrast enhancement. PAS-cytochemistry at EM-level was performed at RT according to standard protocols [49] as previously described [3]: periodic acid (PA: 60 min), thiocarbohydrazide (TCH: 120 min), silver proteinate (SP: 30 min). Note that SP-treatment at RT worked only with quite old SP batches (>30-year-old). Actual batches require heating not to fail, i.e., incubation at +50 °C for 30 min [50] yielded good results.

(e)Tokuyasu immuno-EM [20]

For immuno-gold labeling of thawed cryo-sections samples were processed according to state-of-the-art standards [21,46], with minor modifications. In particular, we used our simplified version [10] of so-called “formaldehyde (FA) fixation at variable pH [51]”. This comprises a short post-fixation step with highly alkaline FA, carried out after normal initial sample fixation with almost neutral FA. Thereby, ultrastructural preservation considerably improves without compromising antigenicity [10], as it might be the case when fixatives contain GA. In brief, biopsies and enteroids from patient #1 were pre-fixed at RT for 30 min to 3 h with FA (4% (*w*/*v*); pH 7.2) followed by 25–30 min post-fixation at RT with 4% FA in 0.1 M sodium borate buffer (38.14 g/L sodium tetraborate decahydrate; adjusted with NaOH to pH 11). Samples were transiently stored in 1–4% almost neutral FA at +4 °C. Subsequently enteroids were rinsed with 0.1M phosphate buffer, suspended in 10–12% (*w*/*v*) warm gelatin, gently centrifuged and finally solidified on ice (note that gelatin embedding is not mandatory for tissue). Very small cubes (side length <1 mm) were cut from the gelatinized enteroids or from naked biopsy tissue, cryo-protected with 2.3 M (787.29 mg/mL) sucrose (in 0.1 M phosphate buffer; overnight at +4 °C), mounted on rivet-like aluminum stubs of a cryo-ultramicrotome, and frozen at −90 °C in the microtome’s cryo-chamber (FCS attached to Ultracut S, both from LEICA). Approximately 100 nm-thin sections were cut at about −110 °C by using a DIATOME cryo immuno diamond knife (from Diatome), picked up in a 1:1 mixture of methyl cellulose (2% *v*/*v*) and 2.3 M sucrose and stored at +4 °C for immuno-gold-labeling. Indirect single or double immuno-gold labeling was performed as described [10,46] with the following primary antibodies (dilutions depend on secondary antibody-conjugates used): Rab11 (1:100, #71-5300, Life Technologies), Rab11a (1:100; #VU57: Ref. [52]), Rab8a (1:50; Ref. [53]), Syntaxin3 (1:30, #ab133750, Abcam), NHE3 (1:100-1:400, #HPA036669, Sigma), DPPI4 (1:10, #AF1180, R & D Systems), CFTR (1:25-1:100, #596mAB: Ref. [54]), LC3 (1:500–1200, #PM036, MBL Intl.), p62/SQSTM1 (1:30, ## PROGEN), LAMP1 (1:10, #H4A3, DSHB), Cathepsin D (1:50-1:200, Ab2, Calbiochem), BSEP (1:10–1:60, #sc-74500, Santa Cruz), MRP2 (1:20, #sc-59608, Santa Cruz), TGN46 (1:40, #AHP500GT), Giantin (1:100, #24586 Abcam), PDI (1:30, clone 1D3, #ADI-SAP-891-D, Enzo), M6PR (1:50, #2733, Abcam). The specificity of the used antibodies or their suitability for EM-immuno-labeling was shown in previous studies [10,27,54] or was validated here (see below Section 2.5. and Appendix Ab). Bound primary antibodies were visualized by relevant secondary antibodies (IgG molecules) conjugated to 5–20 nm colloidal gold (from British Biocell International, Cardiff, UK. or AURION, Wageningen, The Netherlands) or by secondary antibodies (Fab’ fragments) conjugated to Nanogold™ (from Nanoprobes). 1.4 nm-small Nanogold™ conjugates, and optionally 5nm colloidal gold conjugates were subjected to silver enhancement (HQ-Silver™, Nanoprobes).

#### 2.4.3. Electron Microscopy and Electron Tomography for 2D and 3D Analyses, Respectively

Ultrastructural analyses of 100 nm-thin sections were performed with at 80 to 100 kV with a CM120 transmission EM (from Philips, Eindhoven, The Netherlands; now Thermo Fisher Scientific). Digital images were taken with a Morada G1 CCD camera (from EMSIS, Münster, Germany). Scanning transmission EM-tomography was performed with a TECNAI T20G2, equipped with a HAADF detector (all from FEI, now: Thermo Fisher Scientific) at 200 kV. Tomograms were obtained through dual-tilt series from ≈300 nm- resin sections and data were reconstructed and modeled using IMOD software version 4.9 (University of Colorado, Boulder, CO, USA) essentially as described [27]. Software used if not specified elsewhere: GNU Image Manipulation Program version 2.8.10 (GIMP) open-source software, ImageJ, version 1.49a software (National Institutes of Health, Bethesda, MD, USA), Adobe Photoshop CS6 and V.9, Adobe Illustrator CS6, iTEM-analySIS five software (from EMSIS). Image processing was optionally carried out for adjustment of contrast and brightness, grey scale modification, sharpening, median and high-pass filtering. Length measurements were performed with iTEM-tools as described [55].

### 2.5. Cell Culture and Immunoblot Analysis to Validate Anti-CFTR Monoclonal Antibody #596

Cystic fibrosis transmembrane conductance regulator (CFTR) plasmid constructs were a kind gift from Dr. Arlinet Kierbel [Universidad Nacional de San Martín, Buenos Aires, Argentina [56]). HeLa cells stably expressing green fluorescent protein (GFP)-tagged CFTR or CFTR-F508Δ were cultured at +37 °C, 5% CO_2_ in DMEM containing 4 mM (5.84 mg/mL) L-glutamine, 4500 mg/L glucose, 1 mM (0.1 mg/mL) sodium pyruvate, 1500 mg/L bicarbonate, 10% fetal bovine serum and 1% Pen/Strep (5000 U Penicillin and 5000 μg/mL Streptomycin). CFTR-GFP overexpressing clones were selected in medium containing 2 μg/mL puromycin. For immunoblot analysis cells were washed in ice-cold PBS and lysed in a buffer containing 1% NP-40, 150 mM (8.7 mg/mL) NaCl, 50 mM (6.05 mg/mL) Tris, 1 mM (0.38 mg/mL) EGTA, pH 7.4, and a protease inhibitors mix on ice. Protein concentration was determined by Bradford Assay and 12 μg of total proteins were separated by a 7.5% SDS-PAGE. Samples were transferred to a PVDF membrane and the membrane was blocked in a 5% BSA Cohn fraction V-containing 1% TBS-T buffer. CFTR was detected by mouse monoclonal antibody (clone #596 anti-CFTR antibody Ref. [57] diluted 1:1000) overnight at +4 °C. Anti-beta-tubulin (clone E7, #AB2315513, DSHB) was used for normalization of protein amounts. Secondary HRP-conjugated antibody was detected by enhanced chemiluminescence.

### 2.6. Study Approval

Written informed consent from the patients and/or their parents was obtained and biopsies from duodenum and liver were obtained. Written informed consent for testing of MVID genes or genome-wide testing was obtained following genetic counseling. The study was approved by the respective Ethics committees of the universities involved (vote-nr. 1029/2017) according to the principles outlined by the Helsinki declaration. Blood samples were obtained for genetic analyses.

## 3. Results

### 3.1. Clinical Data—Patient Description

Diagnostic biopsies from three patients with MYO5B mutations were studied, as well as control biopsies from patients with unrelated disorders. For IF we analyzed duodenal biopsies from one patient with functional dyspepsia and one biopsy from a healthy liver designated for transplantation. For EM we analyzed duodenal samples from 2 patients with functional dyspepsia (one of them shown in Appendix A) and liver samples from one patient with Friedreich’s ataxia (shown in respective text figures), from one patient with alpha-1 antitrypsin deficiency (not shown), and from a healthy donor liver. All three patients MYO5B mutations with were genetically described recently [44].

Patient #1 is 35-year-old Caucasian male with compound heterozygous MYO5B-mutations: c.242A>G (p.His81Arg) and c.4798C>T (Gln1600*) [44]. Parental consanguinity is not known. Disease onset was at 3 months of age. Diarrhea and cholestasis was reported during infancy. Long-term parenteral alimentation was required but the patient could be weaned over the years. Initially, Byler’s disease was suspected and biliary diversion procedure was carried out at the age of 8 years. Cholestasis resolved after external biliary diversion. Since adolescence, diarrhea occurs only periodically and is well treatable. The patient reported sustained intestinal autonomy during the last follow-up.

Patient #2 is a 12-year-old Arab female, daughter of consanguineous parents, with homozygous MYO5B-mutations: c1669G>T (p.Val557Leu) [44]. Disease onset was reported at 11 months of age with diarrhea and cholestasis. Intermittent parenteral nutrition was required to treat growth failure. External biliary diversion was carried out and later modified to an internal biliodigestive diversion. One of her two brothers carries the same mutations and suffers from similar symptoms (patient #3). Since three years, the patient requires only partial parenteral nutrition. Additionally, she suffered from short stature, hyperparathyroidism, nephrocalcinosis and grade 2 vesicoureteral reflux and concomitant recurrent urinary tract infections. During the last follow-up with 11 years, a good general condition was reported, yet mild persistent cholestatic pruritus was treated with rifampin and colestyramine.

Patient #3, the brother of patient #2, is a 10-year-old Arab male. He carries the same mutations: c1669G>T (p.Val557Leu) [44]. Disease onset was reported at 11 months of age with cholestasis and only intermittent diarrhea. Internal biliodigestive diversion was performed at 2 years of age. No parenteral nutrition is required. Additionally, he initially suffered from short stature, microcephaly and asthma. During the last follow-up with 9 years, a good general condition was reported with height (−1.8 z) and weight (−1.6 z) within the normal limits (using European growth charts [58] and including parental stature). Mild cholestatic pruritus remains and is treated with rifampin.

### 3.2. Fluorescence Microcopy of Duodenal Biopsies

In all three patients intestinal symptoms such as diarrhea and dependence on parenteral nutrition were mild and transient, which is rather unusual for MYO5B-MVID enteropathy. Instead, MYO5B-PFIC and treatment of cholestasis symptoms was evident. To further investigate the patients’ enterocyte morphology, we performed state-of-the-art laser scanning confocal fluorescence microscopy [10,11] on standard cryostat sections of duodenal biopsies. Gross morphology of enterocytes and specifically brush border was not altered as illustrated, for instance, by actin staining (Figure 1a–d). No microvillus inclusions were identified by light microscopy, which is not unusual: those structures are rare or absent in certain MVID cases [10,59], unlike the constant occurrence of aberrant, subapical PAS-staining [10]. In healthy controls, myo5b localized to the apical brush border. In all three patients, mutated myo5b marked in addition a subapical region (Figure 1a). Rab11, a bona fide marker for recycling endosomes (RE), and the apical SNARE protein syntaxin3 showed a similar, even more pronounced mislocalization pattern (Figure 1b,c) which indicates the subapical clustering of MVID-specific endomembrane compartments, historically interpreted as “secretory granules” [10,28]. The apical transmembrane sodium-hydrogen exchanger 3 (NHE3) as well showed cytoplasmic mislocalization, in contrast to a strictly apical localization in healthy controls (Figure 1d). However, the apical localization of the brush border enzyme dipeptidyl-peptidase-4 (DPP4) appeared normal (Figure 1d). Together, the three patients showed mild clinical symptoms and normal-appearing gross morphology of villi, but the intestinal epithelium marker distribution was identical to biopsies of patients suffering more severely from MVID enteropathy [7,10,11].

### 3.3. Electron Microscopy of Duodenal Biopsies and Enteroids

Subsequently the ultrastructure of the duodenal biopsies from patient #1 was studied in detail. To ease comparison with the published literature we first processed chemically fixed duodenal biopsies from patient #1 according to standard procedures (protocol I; for details see also Appendix A, pp. 9–10). Overall, the enterocytes displayed the full range of MVID hallmarks (Figure 2a). Yet, the brush border was almost intact, with only few locally denuded areas. Compared to controls, however, the microvilli appeared generally shorter: ≈0.5–1 µm, instead of ≤2 µm (mean 840 nm ±18, versus mean 1839 nm ±17; Figure 2a, Appendix A). Furthermore, their rootlets often extended unusually deep into the cell (up to 3 µm). We consistently found regions with abundant lateral interdigitations and widened intercellular spaces, as well as bundles of ectopic microvilli facing laterally. Throughout the epithelium pathognomonic microvillus inclusions were a sporadic feature (Figure 2a, Appendix A). However, we also detected groups of enterocytes that contained multiple typical microvillus inclusions per cell. Microvillus inclusions were located both in the apical cytoplasm and more centrally close to the nucleus, measuring up to ≈5 µm in diameter and being regularly surrounded by terminal web or vesicular and tubular endomembranes (Figure 2a). Masses of similar, more or less branched tubules and (sometimes pleomorphic) vesicles (≈100–300 nm-wide) were seen in the subapical cytoplasm of the vast majority of enterocytes (Figure 2a). Applying the EM-correlate of the PAS-reaction [49] we found all those compartments clearly PAS-positive (Figure 2b). Most of the up to 3 µm-large autophagic and lysosomal organelles were also PAS-positive. Dilated rough endoplasmic reticulum (rER) was found in about 20% of the enterocytes (e.g., Appendix A). Whereas all those subcellular features of enterocytes from MVID-patients were already documented in the original description of this disease [28], the biopsy from patient #1 was characterized by an additional ultrastructural anomaly, to our knowledge not yet reported from other cases. Unlike control samples (Appendix A) more than half of the enterocytes displayed heavily dilated Trans Golgi Network (TGN; Appendix A).

In order to investigate enterocyte morphology in greater detail by more advanced EM techniques, we generated enteroids from duodenal biopsies of patient #1. Those in-vitro samples proved as perfectly apt for cryo-fixation tailored for high-resolution microscopy (protocol II). Their major advantages in this respect were relatively small size, easy handling and amenability. In particular, we subjected live enteroids directly to rapid freezing [19], without any need of aldehyde fixation [60]. In total, we analyzed here ≈15 cryo-fixed enteroids (e.g., Figure 2d) from patient #1. Their ultrastructure preservation and membrane contrast were excellent (Figure 2c–e, Appendix A) and well comparable with relevant data from the literature (e.g., [13,55,61,62,63]). As judged from morphology, we obtained enteroids of different maturation stages. Some enterocytes displayed subcellular characteristics of immature enterocytes forming crypt-like shafts (Appendix A) and others resembled more or less mature villus enterocytes (Figure 2c, Appendix A). In brief, those cryo-fixed enteroids mimicked almost completely the subcellular MVID phenotype of chemically fixed biopsies. Most of the features were identical: (i) PAS-positive vesiculo-tubular clusters in the subapical cytoplasm (Figure 2e), (ii) masses of enlarged degradative organelles (lysosomes, autophagosomes Figure 2c), (iii) microvillus inclusions (Appendix A), (iv) local brush border denudation (Appendix A), (v) altered dimensions of apical microvilli (Figure 2a, Appendix A) and (locally) their rootlets, (vi) widened intercellular spaces (Appendix A) and local lateral interdigitations (Appendix A), and (vii) dilated ER (not shown). Only a few, slight differences were observed: (i) less densely packed apical microvilli in enteroids, as compared to biopsies, (ii) no bundled ectopic microvilli facing laterally in enteroids, and (iii) normal TGN and Golgi stacks in enteroids (Figure 2c, Appendix A). Control enteroids (5 samples studied) from a patient with an unrelated disorder appeared normal (Appendix A).

The considerably improved fixation quality through rapid freezing of fresh unfixed specimens provided more detailed information on the endomembrane system of enterocytes with mutated MYO5B. Regarding organelles of the biosynthetic pathway, it seems worth mentioning that some of the enteroid enterocytes showed fragmented, swollen rER, appearing as 250–500 nm-wide vesicles, poorly decorated with ribosomes (not shown). Strongly contrasting with the ER patterns and with our observations of inflated TGN in the biopsy samples, none of the countless Golgi stacks and TGN in the cryo-fixed enteroids showed any swelling at all. In fact, the Golgi stacks of enteroids appeared highly active, as judged from their morphology with elaborate cisternae and abundantly associated vesicles, together with the constant presence of distinct, complex TGN—but all those compartments were free of any dilatation (Appendix A).

Defects in apical cargo transport are now considered as the major pathological mechanism underlying MVID [5,11,59,64]. Our previous studies on tissue samples and relevant cell culture systems had identified the MVID-specific PAS-positive vesicles and tubules in the subapical area as Rab11-Rab8-compartments. Based on the distribution of these two widely used markers and the morphological appearance we interpreted those vesiculo-tubular clusters as recycling endosomes (RE) of altered size, shape and location [10]. Cryo-fixed enteroids from patient #1 showed those aberrant RE as well (Figure 2c,e and Appendix A). Electron tomography [65] yielded accurate reconstructions and models of their 3D architecture (Figure 2f–h; Appendix A), consistent with our previous 3D data from MVID biopsies and rapidly frozen cell models [5,10,27]. In addition to those MVID-specific, quite large aberrant RE, we now documented in cryo-fixed enteroids other, less conspicuous tubular and vesicular organelles (Appendix A). Slim, ≈60 nm-wide, electron-dense tubules located in the juxtanuclear region, but also laterally (Appendix A). They strongly resembled in size, shape and distribution the Rab11-positive “ITV-tubules” reported from cryo-fixed rat intestinal epithelium (Ref. [63]; and our own unpublished data from mouse), as well as the “narrow (Transferrin Receptor (TfR)-positive) tubules in the pericentriolar area [66]” and TfR-positive tubules in cryo-fixed mouse fibroblasts [55]. Similar, ≈60 nm-wide, electron-dense tubules were also present in our cryo-fixed control enteroids (Appendix A)—thus, there seems little doubt they all represent normal, unaltered RE. The apical cytoplasm of immature enterocytes in enteroids from MVID patient #1 and the control patient displayed also scattered, small vesicles (≤100 nm-wide) that were the only vesicular organelles seen touching the apical plasma membrane (Appendix A). Those vesicles, presumably undergoing apical exocytosis could represent genuine “secretory granules” in the sense of early EM-reports (e.g., [67], see [10] for further details). Notably, numerous endocytic pits at the plasma membrane were observed only basolaterally, but not apically in enteroids from patient #1, contrasting with apparently normal patterns in control enteroids (not shown).

MVID enterocytes regularly display conspicuous lysosomes and other large compartments that partially contain still identifiable subcellular compounds and apical marker proteins, including apical sodium transporters sodium/glucose cotransporter 1 (SGLT1) and NHE3, for example [68]. According to morphological criteria those latter compartments have been tentatively classified as autophagosomes and autophagolysosomes [28,31], but without formal proof by autophagy markers. Recently, a potential therapeutic approach for MVID patients has been proposed aiming at restoring normal function of these and other essential molecules by redirecting their aberrant routing from catabolic organelles to the brush border [68]. Moreover, giant, but non-autophagic endo/lysosomal organelles affecting mitosis were newly described from MYO5B deficient CaCo-2 cell cultures; whether they possibly correspond to the seemingly similar, huge compartments in the enterocytes of MVID patients remained open [69]. Considering these aspects, a more reliable characterization and tentative classification of the heterogeneous population of catabolic organelles in our patient biopsies and enteroids seemed advisable. The refined structural data obtained from the rapidly frozen enteroids allowed to distinguish three organelle types (Figure 2c, Appendix A): (i) terminal lysosomes of orthodox morphology characterized by a strongly stained matrix plus varying amounts of degraded remnants (Figure 2c, Appendix A), (ii) short-lived autophagosomes, i.e., islets of cytoplasm with obviously intact organelles surrounded by two sheets of membrane bilayer (“double membrane”: Appendix A), and (iii) huge, spherical compartments harboring just partially degraded organelles within a moderately stained matrix, to be classified as autophagolysosomes (Appendix A; including amphisomes) [22,70,71]. Interestingly, a recent study [13] using the same advanced cryo-fixation approach showed similarly structured degradative organelles in the gut epithelium of a V0-ATPase deficient *C. elegans* model, in addition to other subcellular features reminiscent of human MVID.

To complement our morphological findings, we performed comparative immuno-labeling of biopsy and enteroid samples (protocol III). With single and double immuno-gold labeling according to Tokuyasu [20] we chose the most sensitive, though technically demanding variant of respective EM methods [45]. Notably, structural preservation of cryo-sections from the merely FA-fixed biopsy sample was very good. By contrast, the preservation quality of the FA-fixed enteroids was variable, but still sufficient for analyzing biochemical characteristics of enteroids and biopsy (≈40 enteroids analyzed). First, we aimed at further characterization of the huge catabolic organelles. Supplementing and refining previous immuno-EM-findings of lysosomal acid membrane protein 1 (LAMP1) at those organelles [10,72] we succeeded not only with immuno-gold labeling of cathepsin D but also of the autophagy markers LC3 and p62/SQSTM1. Both in biopsy and enteroids from patient #1 these four catabolic markers showed highly differential distribution patterns (e.g., Appendix A). In line with the morphological data from rapidly frozen enteroids the poorly structured compartments were usually positive solely for LC3 or for LC3 plus LAMP1. Moderately stained compartments with heterogeneous contents were labeled by LAMP1, p62/SQSTM1 and cathepsin D. Compartments representing classical terminal lysosomes, i.e., with distinct matrix contents, showed both cathepsin D and LAMP1 label. Similar marker distributions were seen in control biopsies, though catabolic organelles were generally much smaller (i.e., normal-sized: ≤700 nm) and autophagosomes were very rare. Together, we revealed here for the first time through immuno-EM autophagy markers consistently localizing at various subpopulations of the enormous catabolic compartments in the duodenum and respective enteroids from a MVID patient. Further exhaustive, in-depth multiple-labeling analyses of those organelles were beyond the scope of this study. Anti-giantin, a Golgi marker, and anti-TGN46 (Appendix A) yielded specific labeling of the respective compartments, supporting the ultrastructural observation of strongly dilated TGN in about half of the biopsy’s enterocytes from patient #1. Enterocytes with dilated or normal-sized TGN showed obviously normal immuno-gold label of mannose-6-phophate receptor (M6PR) at the membranes of the TGN and its associated vesicles, as did the control samples (not shown).

In accordance with the results of confocal immuno-fluorescence microscopy, immuno-gold label of Rab8a, Rab11/Rab11a, NHE3 and STX3 appeared greatly translocated to the subapical vesicles and tubules both in biopsy and enteroids from patient #1 (e.g., Figure 3a–d, Appendix A). CFTR-immuno-gold distribution as well differed between biopsy from patient #1 and a respective healthy control (Appendix A; see also Appendix A). In the control, distinct label of varying intensity was concentrated at the enterocytes’ brush border, and scattered peripheral vesicles were also moderately CFTR-positive, in line with respective observations by others [73,74]. In the biopsy of patient #1 the CFTR-label was still present at the apical plasma membrane and microvilli, but appeared generally weak, contrasting with clear labeling of the subapical vesiculo-tubular network. Enteroids showed a similar pattern, though less intense (Appendix A). Finally, both biopsy and enteroid samples from this patient also clearly displayed translocated DPP4 immuno-gold label from the apical plasma membrane to the subapical vesicles and tubules (Appendix A). Such partial mislocalization of this enzyme was not obvious in light microscopy (Figure 1d), likely because of the limited optical resolution obtained with the comparatively thick cryostat sections. However, our present immuno-EM findings are consistent with earlier light microscopy reports on enriched DPP4 in the subapical cytoplasm [75]. A respective synopsis of ultrastructural and immuno-EM observations on enteroids and biopsy of patient #1 is shown in Figure 4 and Appendix A.

At first glance duodenal biopsies from patients #2 and #3 showed no ultrastructural alterations after standard processing (protocol I). Repeated examination of several epoxy sample blocks and sections revealed scattered groups of enterocytes containing subapically a considerable amount of PAS-positive material (Appendix A). However, even a cautious, tentative assignment to known organelles was hardly possible, because of the mediocre structure preservation of those samples. This problem was solved by combining chemical fixation (protocol I) with additional stabilization through HPF/FS (protocol II; for details, see the Materials and Methods chapter). The local accumulations of PAS-positive material were now unambiguously identified as the vesiculo-tubular network and enlarged lysosomes; the latter being confined to cells harboring such vesicles and tubules as well (Appendix A). Other MVID hallmarks were not found. Biopsies for immuno-labeling were not available.

### 3.4. Fluorescence Microscopy of Liver Biopsies

Previous light microscopic observations by others showed mislocalization of Rab11 and apical plasma membrane proteins MRP2 or BSEP in MYO5B-PFIC [38,39,41]. Therefore, we performed detailed light microscopic analyses of liver biopsies from patient #1 taken after external biliodigestive diversion and resolvement of clinical cholestasis and from patient #2 taken before biliodigestive diversion. We used again confocal fluorescence microscopy on standard cryostat sections. Gross hepatocyte morphology and apical bile canaliculi was not altered in both patients as illustrated by actin staining (Figure 5, Figure 6, Figure 7 and Figure 8). Nor differed the apical localization of mutated myo5b in both patients from that of wild-type myo5b in healthy controls (Figure 5). Rab11 localized to the bile canaliculus and additionally marked pronounced aberrant subapical clusters in patients #1 and #2 (Figure 6). Similarly, MRP2 staining was found at the bile canaliculus’ membrane and subapically in both patients (Figure 7). BSEP appeared mislocalized in patient #2, but not in #1 (Figure 8). Whether this difference is attributable to a genetic difference of myo5b or due to the applied diversion procedure remains unknown.

### 3.5. Electron Microscopy of Liver Biopsies

So far, only few data have been published on the ultrastructural pathology of MYO5B-PFIC liver tissue or respective cell culture models [41]. Interestingly, hepatocyte polarity seemed unaffected and microvillus inclusions did not occur. However, conspicuous accumulations of still uncharacterized vesicles were observed [41]. We therefore investigated ultrastructure and immuno-cytochemical properties of a liver biopsy from patient #1 by using the same complementary methods as for intestinal biopsies—except for enteroid technology.

Standard processing (protocol I) of the biopsy from patient #1 showed acceptable quality of ultrastructure preservation (Figure 9a), except for the Golgi stacks and TGN that were not unequivocally recognized. Most remarkable was the abundance of ≈100–200 nm-wide vesicles and short tubules (Figure 9a). As judged from morphology those vesicular endomembranes could be smooth ER (sER). Furthermore, prominent, dilated endomembrane cisternae with electron-lucent contents were numerous (Figure 9a). Most of them could represent rER, though attached ribosomes were rarely observed. Structurally different, “swollen membrane sacks” with poorly defined pleomorphic contours were seen in areas normally occupied by the Golgi stacks/TGN (Figure 9a). Control liver samples from 2 patients with unrelated disorders, and one sample from healthy liver designated for transplantation did not show masses of small vesicles and highly inflated cisternae (e.g., Figure 9b).

Immuno-gold labeling (protocol III) was again the method of choice to evaluate our assumptions concerning sER, rER and Golgi stacks in biopsies from patient #1. Immuno-gold label of Protein disulfide isomerase (PDI), an ER marker, located specifically at the bead chain network of vesicles and short tubules, in addition to the majority of large cisternae with clearly contrasted membranes and scarce ribosomes (Figure 9c,d). Those cisternae, now identified as rER, were additionally characterized by quite numerous, about 75 nm-wide coated buds (Figure 9d), likely representing buds at ER-exit sites [24]. Anti-giantin and anti-TGN46 bound to specific parts of the pleomorphic, indistinct “membrane sacks” and associated vesicles (Figure 9e) in the hepatocytes’ apical regions facing the bile canaliculi (“pericanalicular region” in the sense of Ref. [25]). Notably, the here used anti-PDI, anti-giantin or anti-TGN46 antibodies labeled only some of the 120–200 nm-wide, moderately stained vesicles in the pericanalicular region (e.g., Appendix A). This strongly indicated the existence of at least one other vesicle population. PAS-cytochemistry as well yielded differential staining throughout those vesicle clusters (not shown). Considering our immuno-fluorescence data (Figure 6 and Figure 8) and published literature [10,11,25,38,41,42,74], the MYO5B interaction partners Rab11 and Rab8 as well as BSEP were obvious candidates to test by immuno-EM. Unlike control samples, cholestatic hepatocytes showed anti-Rab11/anti-Rab11a immuno-gold particles to be locally enriched in the pericanalicular area. They located at ≈100–200 nm-wide vesicles as well as in their close vicinity (Figure 9f,g, Appendix A). In addition, Rab11 located canonically at the TGN, similar to Rab8. Rab8 label was also found in the apical cell periphery, but appeared in general normal (not shown). Anti-BSEP label (Figure 10a) in the patient biopsy (#1) was very scarce at the apical plasma membrane and microvilli, and moderate in the pericanalicular area of cholestatic hepatocytes, contrasting to its normal distribution [25] in controls (Figure 10b). MRP2 was in general properly locating at the apical plasma membrane, and was only sporadically found in the subapical cytoplasm as well (Figure 10c). Anti-DPP4 immuno-gold-labeling pattern appeared normal, confined to the plasma membrane. Thus, immuno-EM of pericanalicular vesicle clusters can be summarized as follows. The numerous ≈150 nm-wide vesicles located subapically comprised multiple types, of which the most conspicuous and locally most abundant one was Rab11 positive. Further work on those vesicle clusters is needed.

Data available as yet do not allow any correlation of the provisional immuno-cytochemical classification of vesicles with morphological characteristics and/or PAS-reactivity. Bearing this caveat in mind, we nevertheless generated an electron tomographic reconstruction and 3D model of a cholestatic hepatocyt’s subapical region for better illustration. We just distinguished between ≈150 nm-wide vesicles (pooled) and inflated rER cisternae (Figure 10d–f). The above mentioned “hybrid technique” (protocol IV) used here to enhance membrane contrast as obligatory for tomography had an additional positive effect. The otherwise ill-defined Golgi stacks and TGN were at least locally better preserved, allowing their identification solely according to morphological criteria. A synopsis of structural and immuno-cytochemical observations is shown in Figure 11 and Appendix A.

## 4. Discussion

By using advanced, complementary sample preparation and microscopy techniques we comparatively documented here the ultrastructural and immuno-cytochemical phenotypes of a MYO5B-PFIC patient in great detail, who present besides a constant hepatopathy only periodically with clinical intestinal symptoms. Those findings were supplemented by selected data from two other relevant cases.

Both in hepatocytes and enterocytes from patient #1 the biosynthetic compartments appear functional and even hypertrophic. This is deduced from the constant presence of dilated rER with numerous ER-exit buds, enlarged TGN and/or Golgi cisternae, as well abundant sER in hepatocytes—and similarly inflated TGN in enterocytes. Hypertrophic sER is typical for cholestasis [76,77,78], dilated rER is a sign of high metabolic activity or stress. TGN dilatation, however, has to our knowledge only rarely been reported. Virus infection, for example [79,80], is one of the few naturally occurring disease conditions that induce TGN swelling (unlike drug treatment or other experimental interference). We interpret the highly inflated and locally ruptured appearance of the TGN in chemically fixed duodenal biopsies from patient #1 as fixation artifact, though not an unspecific one. This notion is based on the following arguments. Cryo-fixed enteroids from patient #1 are characterized by all well-known MVID hallmarks—but not TGN swelling. By contrast, they show elaborate Golgi stacks and TGN of complex and well-organized architecture, pointing to high metabolic activity. This would fit to previous data [81] indicating undisturbed biosynthesis of brush border constituents in the enterocytes of MVID patients. Interestingly, a first preliminary re-examination of 4 of our previously published MVID cases yielded mixed results. The features observed ranged from completely normal to clearly dilated TGN, although chemical fixation and subsequent processing were in all cases the same. Together, we consider the dilated TGN phenotype not as negligible, unspecific fixation artifact. Instead, it apparently reflects the metabolic status, stress or even the pathological condition of the affected cells. Similar phenomena are known from various tissues and organisms, such as hypertrophic cartilage, ageing keratinocytes or injured retina cells [82,83,84], see also Ref. [85] for more details. Clearly, more work is necessary concerning the possible biological relevance of the here described dilated TGN phenotype in duodenal and liver tissue of patients suffering from MVID, PFIC or both.

In contrast to apparently proper biosynthetic and M6PR pathways, aberrant routing and finally jamming of cargo have been established as cornerstones of MVID pathophysiology [5,11,64,86]. Over the last years, MYO5B, STX3 and STXBP2 have been identified as MVID causing genes [4,11,34] and their interplay in apical exocytosis in enterocytes was elucidated. Interestingly, patients with nonsense mutations presumably merely cause MVID enteropathy, while only patients with missense mutations in MYO5B develop cholestasis [44]. Cholestasis has not been reported from the very few MVID patients with STX3 or STXBP2 mutations so far. This might reflect our limited knowledge on this specific condition. Alternatively, cargo transport towards the bile canaliculus could differ from transport to the apical surface in enterocytes. The latter assumption is supported by the lack of hepatic phenotypes in patients with nonsense mutations in MYO5B. MYO5B knock-out hepatocyte models as well lack a transport phenotype in vitro [41]. Arguably, missense mutations exert a dominant-negative effect on a tightly regulated transport system that might rely on redundant motor proteins in hepatocytes. The fusion protein machinery at the apical pole of hepatocytes remains also to be deciphered, as preliminary observations suggest mechanisms independent of syntaxin3 and syntaxin-binding protein2.

We add here to this puzzle a first data set on the ultrastructure and immunogold distribution of selected markers in the subapical region of hepatocytes from MYO5B-PFIC patient #1. In line with previous light microscopic findings [38,41,42] we observed immuno-gold label of the small GTPase Rab11, a bona fide marker for RE and/or apical exocytosis in epithelial cells, partially mislocalized to vesicle clusters surrounding the bile canaliculi. Such patterns of Rab11-positive subapical vesicles in hepatocytes with MYO5B mutations resemble to a certain extent the aberrant vesiculo-tubular Rab11-Rab8-harboring compartments in enterocytes of MVID patients. However, there are clear ultrastructural differences. The respective organelles in enterocytes form anastomosing networks, are wider in diameter and constantly strongly PAS-positive [10]. The Rab11-vesicles identified here in MYO5B-PFIC hepatocytes are individual vesicles of smaller size, and at the best moderately PAS-positive. In addition, specific enrichment of Rab8 in those vesicle clusters could not be ascertained so far. Thus, further work on those mixed vesicle populations is needed, since it still remains unclear whether the moderate, locally observed subapical enrichment of BSEP and MRP2 corresponds to the more conspicuous Rab11-positive vesicles. BSEP and MRP2 mislocalization was previously reported from MYO5B-PFIC tissue samples and in vitro models [38,39,41,42]. In liver biopsies of patient #1 and #2 immuno-fluorescence microscopy showed mislocalization of BSEP to be quite distinct, that of MRP2 only moderate. As the low GGT PFIC 2 is based on disrupted bile salt transport due to mutations in ABCB11/BSEP itself, aberrant BSEP transport due to mutations in MYO5B explain the clinical phenotype well. On clinical and pathophysiological grounds, MYO5B-PFIC is correctly categorized as low GGT PFIC. Mislocalization of MRP2 could only add to PFIC symptoms and might lead to aggravation of biochemical serum alterations in MYO5B-PFIC patients. Interestingly, biliodigestive diversion procedure seems effective in MYO5B-PFIC as all three reported patients did clinically benefit. This suggests that ileal bile acid uptake remains functional in MVID patients and disruption of the enterohepatic circulation poses a relief to cholestasis. Speculatively, upcoming intestinal bile salt uptake inhibitors might be effective in MYO5B-PFIC patients as well. But first, further studies on Ileal sodium/bile acid cotransporter (SLC10A2) trafficking in MVID are necessary.

In all three patients studied, intestinal symptoms were severe in the first months to years of life followed by a cessation. In patient #1 periodical episodes of diarrhea are noted, but limited in duration which seems surprising, given the pronounced ultrastructural phenotype of his enterocytes. Duodenal biopsies and enteroid cultures displayed almost every hallmark of MVID—only the apical brush border remained intact and generally not eroded. What might then account for intestinal autonomy in MYO5B-PFIC patients? Arguably, missense mutations could show some remaining function, or redundant pathways were exploited by enterocytes. However, this hypothesis was not proven experimentally so far. One possible explanation might be the increase of absorptive surface and maturation of the intestine. If patients manage to survive throughout the first years of life by the help of parenteral nutrition, intestinal growth could have caught up to intestinal autonomy. Recurrent episodes of diarrhea might then just reflect temporary failure of this fragile equilibrium.

Over the last years, the use of enteroid cultures has rapidly set foot in the field of congenital enteropathies [4,11,12,87]. Intestinal organoids, so-called enteroids, were established from patients and respective mouse models, as they allow detailed studies on enterocyte morphology and function overcoming limited availability of tissue specimens. Enteroids used in this study recapitulated the cytological MVID hallmarks, both in terms of subcellular architecture and biochemical marker expression and distribution. This included, for example, obviously proper expression and (aberrant) localization of the essential brush border transporter NHE3. To our knowledge, our work presents for the first time a comprehensive EM-analysis of human organoids that were cryo-fixed in their native state, without any preceding chemical fixation. As detailed above, this advanced specimen preparation methodology together with state-of-the-art microscopy and immuno-EM-approaches yielded more detailed insight into the complex endomembrane compartments of enterocytes with mutated MYO5B. Moreover, enteroids have previously proven as apt for more than just static structural analyses: advanced live cell microscopy performed on FHL5/MVID enteroids provided a wealth of information on the dynamics of microvillus inclusions, their formation and dissolution [12]. Hepatocytic organoids have been presented only recently [14,41] and therapeutic research will greatly benefit from respective platforms. Together, we believe our study provides further evidence for the reliability of enteroids. Thus, such model systems are likely capable to mimic diseases in vitro to a great extent, thereby serving as efficient tools in personalized medicine [59].

Finally, we recommend a revival of formaldehyde (FA) as versatile, almost universally usable primary fixative for any kind of EM-analysis of biopsy material. We are well aware of contradicting here worldwide established practice (see current pathology guidelines [1]) and truly respect the excellent results obtained with GA [88] in the long tradition of biomedical EM. Especially in the case of rare material from orphan diseases and/or minute sample size initial fixation with buffered FA (diluted from paraformaldehyde) leaves much more opportunities for subsequent analyses (including GA-postfixation). Details are explained in the Materials and Methods chapter. Concentrations higher than ≈0.1–0.2% (*v*/*v*) of GA irreversibly destroy antigenicity and/or other reactive sites [45], and antigen retrieval for EM-studies is still in its infancy [89], except for few favorable applications [90]. Instead, short post-fixation with alkaline formaldehyde (Ref. [10], modified after Ref. [51]) considerably improves ultrastructure preservation without compromising antigenicity.

## Figures and Tables

**Figure 1 jcm-10-01901-f001:**
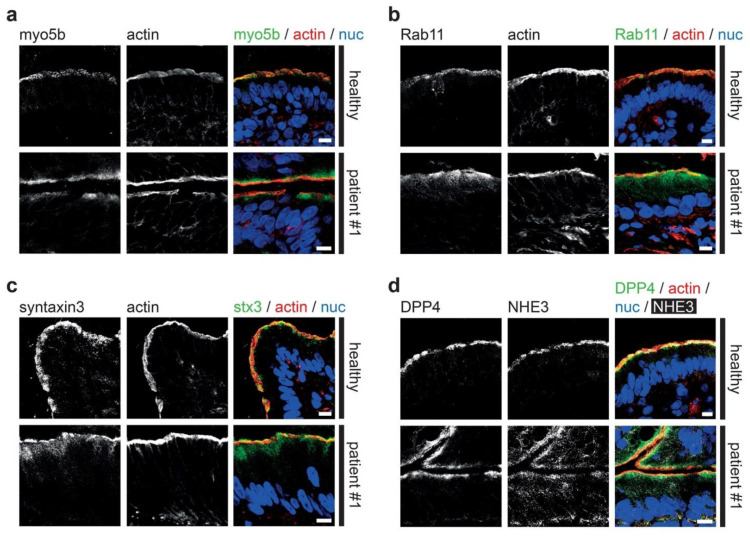
**Aberrant localization of apical proteins in MYO5B-PFIC enterocytes.** Laser-scanning immuno-fluorescence micrographs of duodenal enterocytes of patient #1 show (**a**) apical and aberrant, subapical localization of mutated myo5b, subapical accumulation of Rab11 (**b**) and syntaxin3 (**c**), as well as cytoplasmic mislocalization of NHE3 (**d**) in comparison to controls. DPP4 localization (**d**) seems unaltered. Brush border microvilli indicated by actin staining; scale bars = 10 µm.

**Figure 2 jcm-10-01901-f002:**
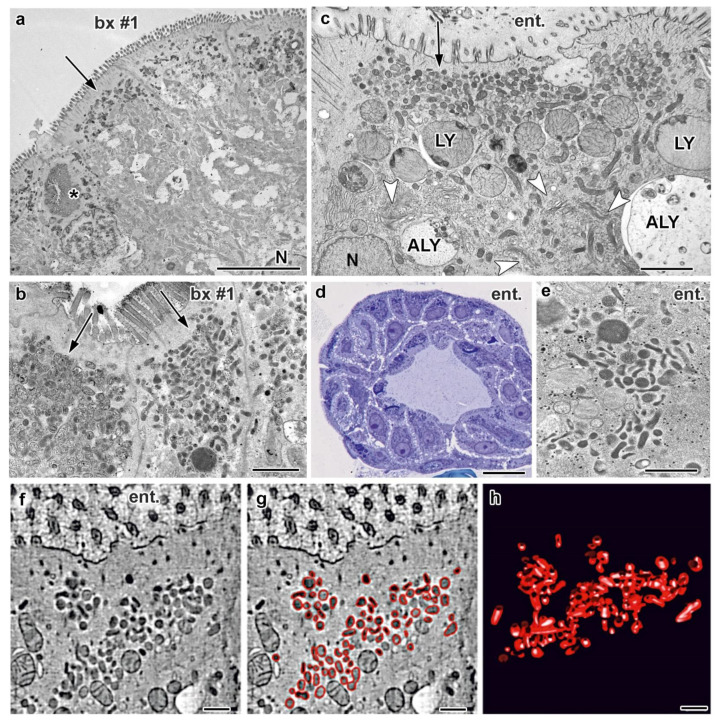
**Ultrastructural MVID-phenotype in duodenal biopsy (bx #1) and enteroid samples (ent.) from MYO5B-PFIC patient #1.** (**a**) Villus enterocytes from chemically fixed biopsy showing a microvillus inclusion (asterisk) and subapical clusters of aberrant, dark recycling endosomes (RE: arrows) specific for this disease [10]; N = nucleus; scale bar = 5 μm. (**b**) PAS-cytochemistry at EM-level detects strongly reactive compounds in aberrant RE; scale bar = 1 μm. (**c**) Cryo-fixed enteroid displaying MVID-specific, subapical RE clusters, as well as huge lysosomes (LY) and autophagolysosomes (ALY), but normal Golgi/TGN (arrow-heads); scale bar = 2 μm. (**d**) Light micrograph of cryo-fixed enteroid highlights the high vitality of those enterocytes; scale bar = 20 μm. (**e**) Distinct PAS-reaction of subapical RE in cryo-fixed enteroid; scale bar = 500 nm. (**f**–**h**) Electron tomographic reconstruction of 3D-architecture of subapical RE in cryo-fixed enteroids. 2D slice (**f**) from an electron tomographic reconstruction with red contour lining (**g**) and the resulting 3D model (**h**) of those organelles (see also Supplementary Movie M1); scale bars = 500 nm.

**Figure 3 jcm-10-01901-f003:**
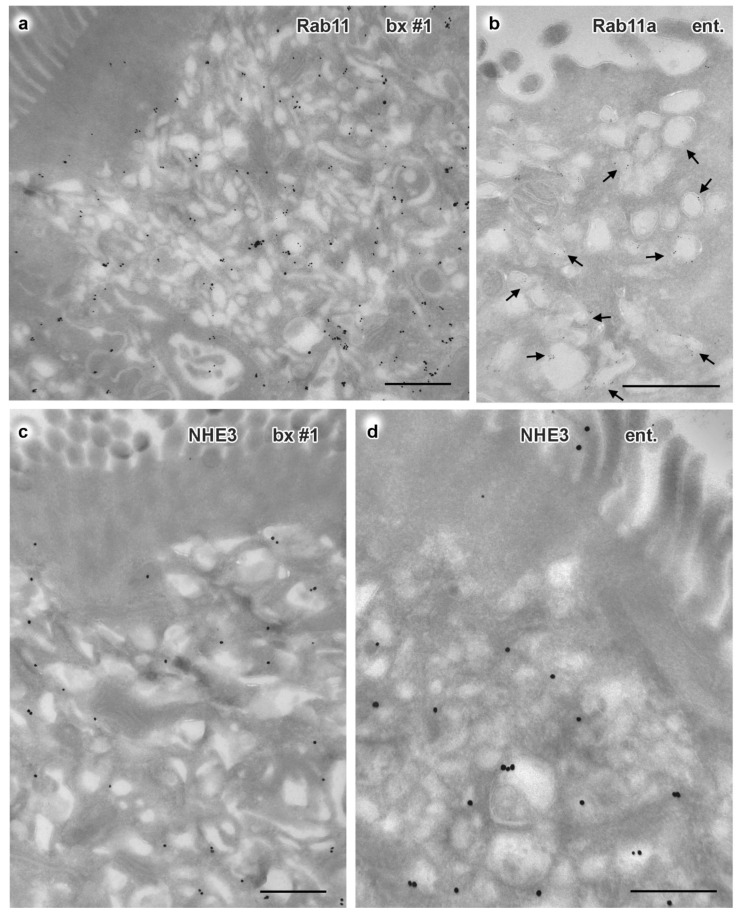
**Immuno-gold EM of cryo-sections from formaldehyde-fixed duodenal biopsy and enteroid samples from MYO5B-PFIC patient #1**. Corresponding mislocalization patterns of apical markers to aberrant subapical RE are evident in biopsy #1 (**a**,**c**) and enteroid (**b**,**d**); all scale bars = 500 nm. (**a**) Rab11 in biopsy visualized by 1.4nm Nanogold plus silver enhancement. (**b**) Rab11a in enteroid visualized by 5nm colloidal gold (arrows). (**c**) NHE3 in biopsy (enhanced Nanogold), (**d**) NHE3 in enteroid (enhanced Nanogold).

**Figure 4 jcm-10-01901-f004:**
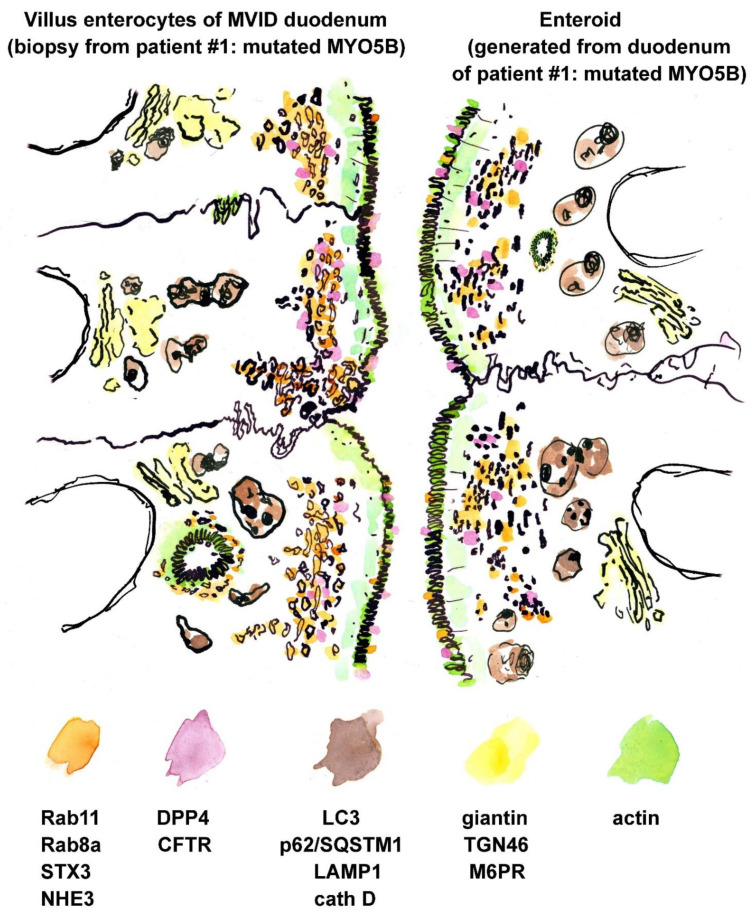
**Synoptic illustration of immuno-gold and ultrastructural characteristics of duodenal biopsies and enteroids from MYO5B-PFIC patient #1**. Compare this particular case with a respective synopsis of multiple MVID cases in Figure 2 of Ref. [10]. Drawing almost to scale. See also Appendix A.

**Figure 5 jcm-10-01901-f005:**
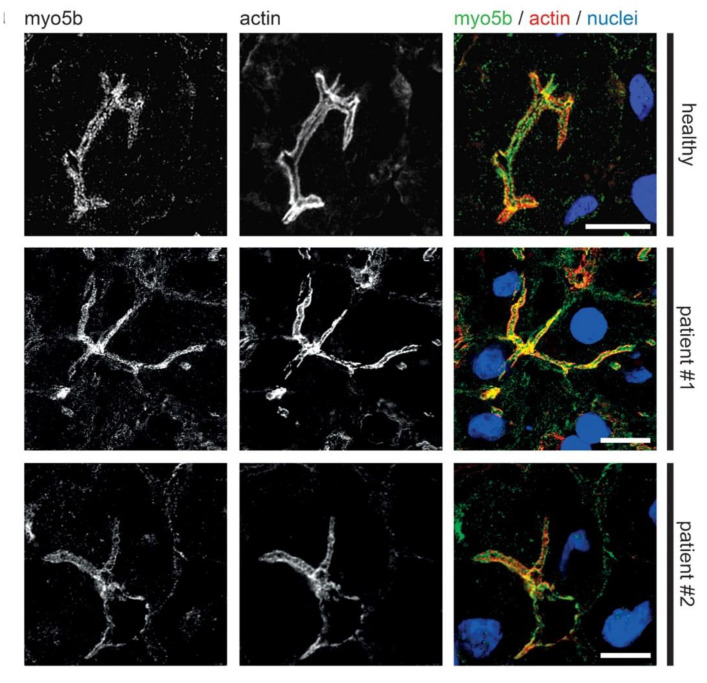
**Localization of myo5b in MYO5B-PFIC hepatocytes.** Laser-scanning immuno-fluorescence micrographs (IF) of hepatocytes from patient #1 and #2 show regular localization of mutated myo5b to the apical plasma membrane compared to healthy controls. Bile canaliculus with microvilli indicated by actin staining; scale bars = 5 µm.

**Figure 6 jcm-10-01901-f006:**
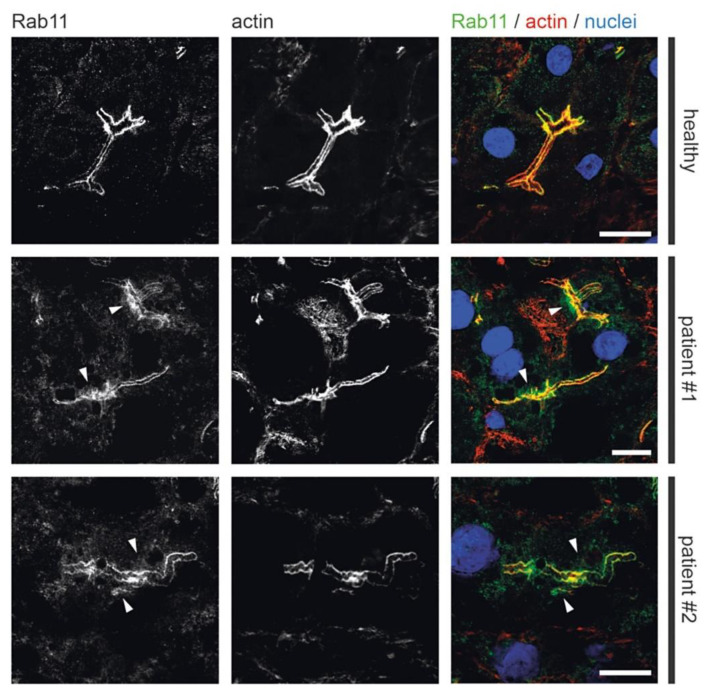
**Localization of Rab11 in MYO5B-PFIC hepatocytes**. IF of hepatocytes from patient #1 and #2 show subapical accumulation of Rab11 (arrow-heads) compared to healthy controls. Bile canaliculus with microvilli indicated by actin staining; scale bars = 5 µm.

**Figure 7 jcm-10-01901-f007:**
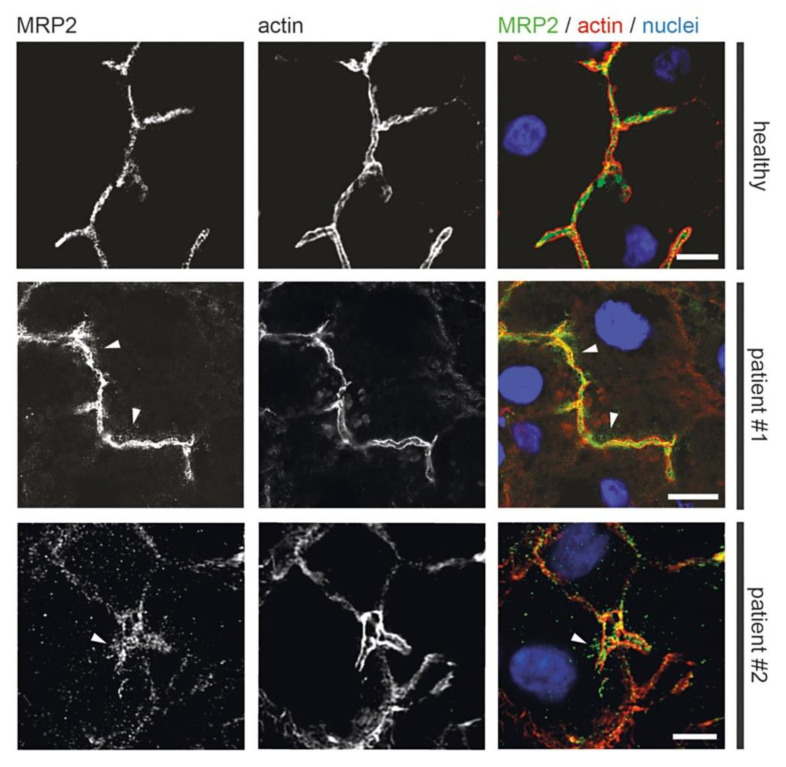
**Aberrant localization of apical proteins in MYO5B-PFIC hepatocytes.** IF of hepatocytes of patient #1 and #2 show subapical accumulation of MRP2 (arrow-heads) in both patients compared to healthy controls. Bile canaliculus with microvilli indicated by actin staining; scale bars = 5 µm.

**Figure 8 jcm-10-01901-f008:**
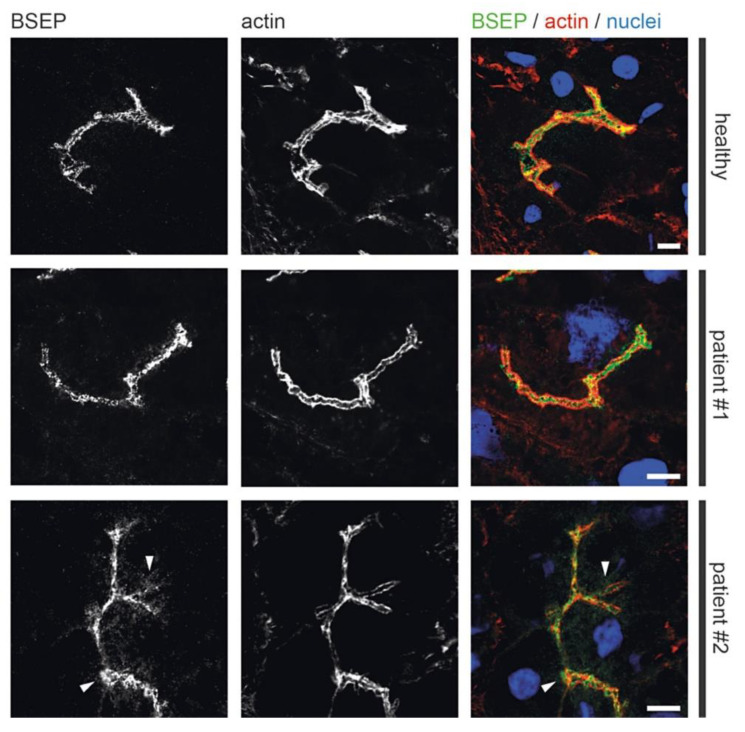
**Aberrant localization of apical proteins in MYO5B-PFIC hepatocytes**. IF of hepatocytes of patient #1 and #2 show subapical accumulation of BSEP in patient #2 (arrow-heads), but not in patient #1 and in healthy controls. Bile canaliculus with microvilli indicated by actin staining; scale bars = 5 µm.

**Figure 9 jcm-10-01901-f009:**
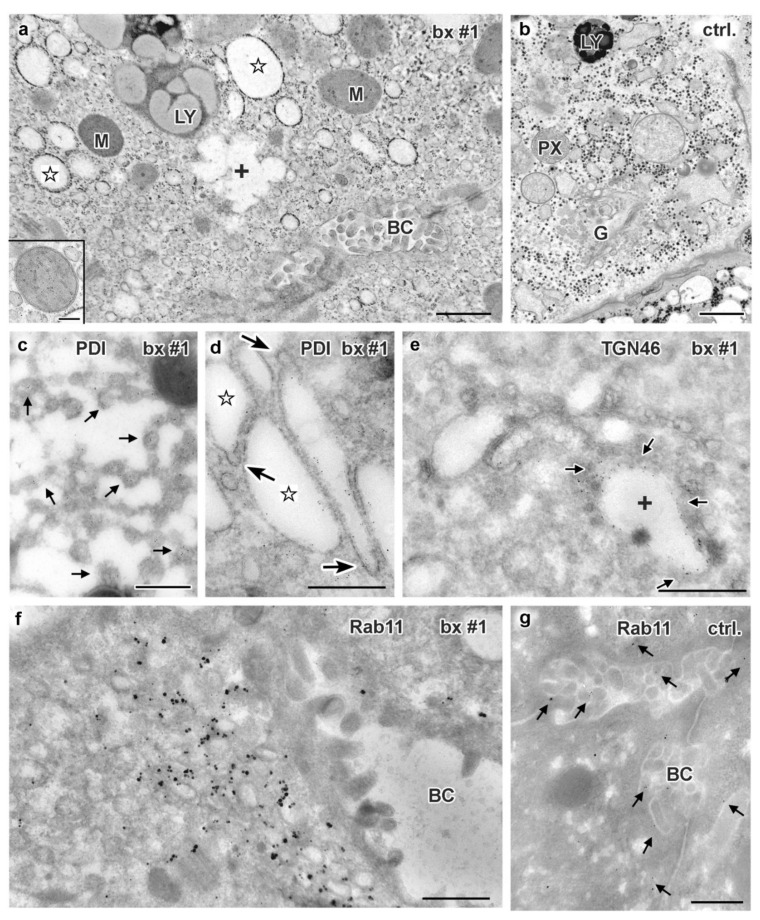
**Ultrastructure and immuno-gold EM of aldehyde-fixed liver biopsies from MYO5B-PFIC patient #1 (a,c–f) and control (b,g).** (**a**) Endomembrane compartments distinguished in MYO5B-PFIC sample comprise large, ribosome-decorated unstained cisternae (marked by stars), ill-defined “swollen membrane sacks” (marked by a cross: +), abundant small vesicles and lysosomes (LY). Mitochondria (M) appear normal, peroxisomes and locally occurring large lipid droplets (indicating steatosis) are not shown in this frame of apical hepatocyte area facing bile canaliculus (BC); scale bar = 1 μm. Insert shows a mitochondrion at higher magnification; scale bar = 200 nm. (**b**) Control biopsy with normal, well-organized Golgi stack/TGN (G). Note LY and peroxisome (PX) nearby; Scale bar = 1 μm (**c**) ER-marker PDI labels specifically 100–200 nm-wide, small vesicles/tubules (arrows) identifying sER; scale bar = 500 nm. (**d**) PDI labelling of ribosome decorated cisternae (stars), thus, rER, that display numerous 75 nm-wide buds (arrows), likely at ER-exit sites; scale bar = 500 nm. (**e**) Anti-TGN46 label (arrows) at cisternae (cross) and associated vesicles identifies “swollen membrane sacks” as dilated TGN; scale bar = 1 μm. (**f**) Rab 11 label clusters locally at, or in close vicinity of ≈150 nm-wide vesicles in the hepatocytes close to the bile canaliculus (BC); scale bar = 500 nm. (**g**) Normal Rab11 distribution (arrows) in control hepatocytes; scale bar = 500 nm.

**Figure 10 jcm-10-01901-f010:**
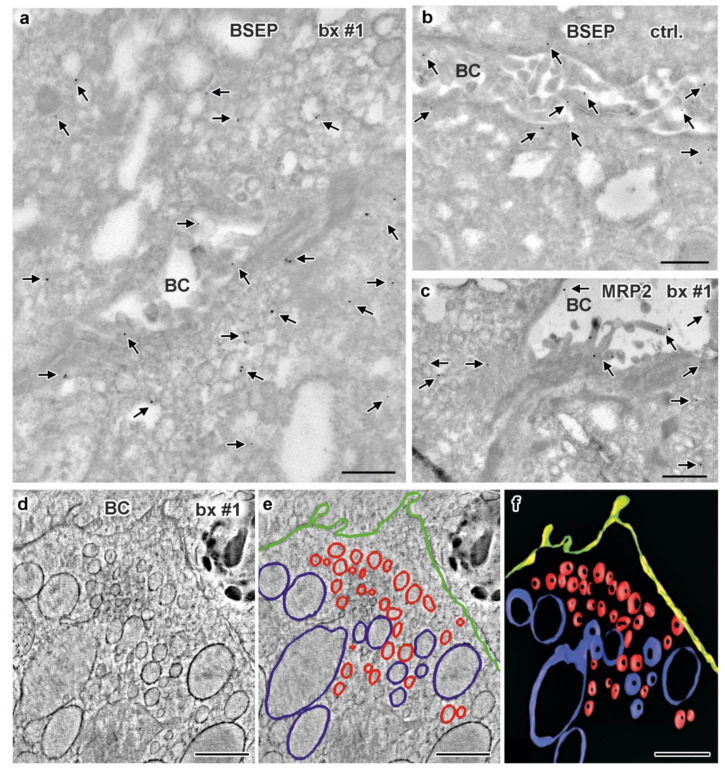
**Immuno-gold EM of cryo-sections and 3D reconstruction of subapical vesicle clusters in hepatocytes of biopsy from MYO5B-PFIC patient #1 (a,c–f) plus labeling control (b).** (**a**) BSEP immuno-gold label (arrows) appears weak at the apical plasma membrane but abnormally enriched in the cytoplasm and/or at vesicles around the bile canaliculus; scale bar = 500 nm. (**b**) Bona fide normal BSEP distribution (arrows) concentrated at the plasma membrane of control hepatocytes; scale bar = 500 nm. (**c**) Very moderate, local mislocalization of MRP2 (arrows) in biopsy of patient #1; scale bar = 500 nm. (**d**–**f**) Electron tomographic reconstruction of 3D-architecture of subapical vesicles (plus rER cisternae) in hepatocytes of aldehyde-fixed, HPF/FS sample from patient #1. 2D slice (**d**) from an electron tomographic reconstruction with contour lining (**e**) of vesicles (red), rER (blue), plasma membrane (green) and the resulting 3D model (**f**); scale bars = 600 nm.

**Figure 11 jcm-10-01901-f011:**
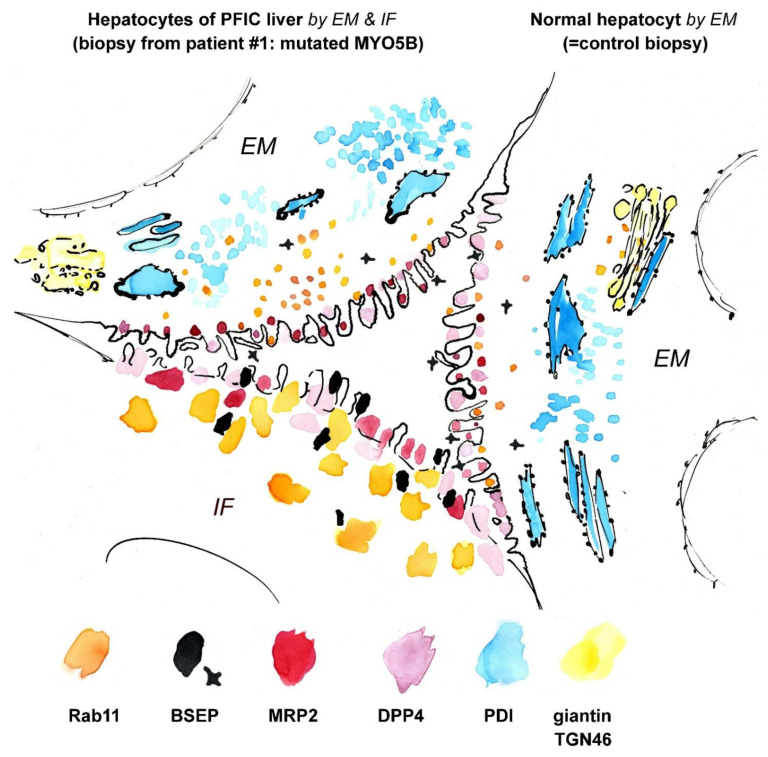
**Synoptic view on a virtual (“compound”) bile canaliculus.** The illustration compares the immuno-labeling patterns of cholestatic hepatocytes from patient #1 (with mutated MYO5B; shown on the left) with those of normal hepatocytes (shown on the right). Immuno-EM (EM) provided more detailed information in terms of structure and the precision of marker localization than immuno-fluorescence (IF). Partly mislocalized Rab11 and BSEP label is seen in cholestatic hepatocytes, whereas DPP4 and MRP2 are located normally. Drawing roughly to scale. See also Appendix A.

## Data Availability

The data presented in this study are available on request from the corresponding authors.

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
