# Peer review of "Advanced Microscopy for Liver and Gut Ultrastructural Pathology in Patients with MVID and PFIC Caused by MYO5B Mutations"

_jcm, 2021, doi:10.3390/jcm10091901_

Round 1

Reviewer 1 Report

Review:

The manuscript of Hess et al. analyzed the ultrastructural phenotypes of the cells in gut and liver in MYO5B-PFIC patients with mutations in the actin motor protein myosinVb causing the microvillus inclusion disease (MVID)

The authors characterized the ultrastructure of different types of large catabolic organelles in enterocytes as well as the distribution of autophagic and lysosomal marker proteins within them.
In addition to morphological alterations, many proteins for the apical cargo transport were similarly mislocalized in enterocytes as well as in duodenal organoids (enteroids) which the authors cultured from patient biopsies.
Liver biopsies of MYO5B-PFIC patients showed a strong mislocalization of Rab11 and a more week one for bile canalicular membrane proteins. Comparable to the accumulation of recycling endosomes in enterocytes of MVID patients Rab11-positive vesicles clustered locally around bile canaliculi.

To achieve these new and interesting results, the authors used and improved several advanced electron microscopic methods like cryo-fixation, freeze-substitution, immuno-gold labeling, electron tomography and immuno-fluorescence microscopy.

I have some comments to improve the paper:

In general: You should keep in mind that most of the ultrastructural and immunolabeling results are found in the well characterized gut and liver cells of only one patient. This description is necessary and may lead to a better understanding of the MVID disease and possibly help to develop a therapy. But the authors should reread the text in order to avoid any general statements that may not be claimed after the investigation of only one extraordinary case. 21: We thoroughly characterized here the ultrastructural and immuno-cytochemical phenotype of hepatocytes and duodenal enterocytes from MYO5B-PFIC patients, who… It was in most cases only done for one patient. See line 594: “Biopsies for immunolabeling were not available. (..for patient #2 and #3)

50, 232, 255, 302, 452: unnecessarily nested brackets

220: “completely lost/masked their…” - Please write clear it is too unspecific.

222f: the method description should state clearly what kind of FA fixative was used; neutral = 7.2 pH?

226: 8% “aquous”? A. dest?

243: “an ascending acetone series is stated for dehydration although protocol I..” Suppl. methods mention EtOH?

244: the phrase “absolutely unnecessary” should be avoided to not be misinterpreted as an intermediate would be unnecessary at all (in every case)

258: “resins based on Epon and/or Araldite, but not Spurr’s resin.” - Unclear which resin did you use?

265: “staining solution (a lanthanide cocktail)”?

301, 305, 413, 417, 494: and/or – Unclear, specify

311: with at… typo, kv should be kV

316: “semi-thick” means semithin?

333: ice-cold 1x PBS?

351-352: “plus controls from patients with unrelated disorders” - Please specify, how many control patients were analysed and which disorders did they have.

Fig1a: the label “myo5B” should be lowercase “b”?

Fig2: label “ent.”? “bx”? – Description is missing in the caption, confusion possible: “ent.” enterocyte? enteroid?

411: …shorter (≈0.5-1μm, instead of ≤2μm (Figure 2a, Figure S2a): quantification, vague <=, closing bracket

415: “occur as well, but at variable frequency”? - Which frequency?

422: “(auto)lysosomal” – Unclear, specify

423: “about one fifth of the enterocytes” - How many, tell numbers?

427: the new description of the heavily dilated Trans Golgi Network is depicted in a very small subfigure (2b) and therefore hardly recognizable, unfortunately.

434: “normal, dark recycling endosomes (RE) specific for this disease” - the images Fig. 2a, c, … show no label for (RE) in the figures? 

435: “inflated Golgi/TGN marked by a cross” please enlarge the image in Fig 2b, is too small the structure is not visible. I think it is worth to show. See comment for line 427.

449  >15 – How many?

454: ≙  asci character uncommon?

455: “enteroids mimicked almost completely the subcellular MVID phenotype…” what was different and what similar?

475: “interpreted the vesicles/tubules as abnormal recycling endosomes (RE: (9)).” compared to
484: “there seems little doubt they all represent normal, unaltered RE.”  ?

483: add a space before (61)    

503: bona fide, meaning "in good faith", which is often used as an adjective to mean "genuine".
I would replace in the whole text “bona fide” with real, authentic or genuine.

503: autophago(lyso)somes? – Unclear, specify

516: cytoplasm with/without +/− intact organelles? – Confusing, unclear, specify

519: autophago(lyso)somes? – Unclear, specify

530: (>40 enteroids? - How many, number?

555, 558, … vesicles/tubules? – Unclear, do you mean vesicles and tubules?

570: (immuno-)EM?

Fig. 3: arrows? mark abnormal subapical RE, bx #1, ent.? – Label for RE in the figures? As in Fig. 2 description of “bx” and “ent.” is missing in the caption, arrows in Fig. 3b are not mentioned in the caption. 

585: “method 1” = protocol1?

Figures general: subfigures and scale bars are not exactly aligned, scale bars often with and without white halos

599: omit “electron microscopic” – in the section fluorescence microscopy of liver biopsies

638:  acceptable ultrastructure preservation (Figure 9a)? in which respect?

686: normal mitochondria?? Where are the inner membranes of the mitos?

Fig. 9f the big, empty lumen of the bile canaliculus (BC) is atypical compared to the (BC) in Fig. 9g. Maybe a part of the sinusoid?

646: 2 patients with unrelated disorders, and one sample from healthy liver designated for transplantation … Fig. 9f which of the 3 controls is shown? Description missing

653: network of vesicles/tubules

667:  locating at and/or in close vicinity

Fig. 10 is surrounded by a line box which is not present in all other figures.

699: (plus control) – no 3D data of controls are shown

700:  locally enriched (i.e., mislocalized) in the cytoplasm and/or at vesicles around – really? …not traceable, confusing, unclear.

704:  vesicles/tubules – no tubules are shown

706, 712: rER (blue) – why not “ribosome-decorated unstained cisternae (rC)” of Fig 9a. You may change the label (rC) to (rER). Different names for the same structure. 

713: “hybrid technique” (protocol IV) – and/or 2.5% GA ?? Did you apply GA or not?

Fig. 11: synopsis – mixture of control – mut,  LM – EM !! Very confusing for the reader.

728: the ultrastructural and immuno-cytochemical phenotypes of MYO5B-PFIC patients - Data only from one patient, see above.

812: “Enteroids used in this study completely recapitulated the cytological MVID hallmarks” 798: “displayed almost every hallmark of MVID” 409: “displayed the full range of MVID hallmarks” differences are described in 409-430

Author Response

Thank you very much for the quick and carefull evaluation as well as the invaluable hints for improvement. Our point by point reply is attached, and changes in the manuscript are highlighted in yellow.

Reviewer 2 Report

By using state of the art microscopy techniques, Hess et al. describe here a unique case of a patient suffering from MYO5B-PFIC, showing constant hepatopathy and periodical enteric symptoms. The clinical case is clearly exposed and supported with data from two more patients. The samples have been well studied at the ultrastructural level and these data complemented with immune-labeling and fluorescence microscopy for validation and a better characterization. Moreover, the authors provide well-detailed protocols and some clues for a better sample handling, in order to maximized data collection from usually very limited samples. The authors have also derived enteroids from the human sample, which closely reflect the phenotype observed in intestinal patients. Although a number of publications in the recent years have provided with new data in the mechanisms underlying MVID and MYO5B-PFIC pathogenesis, we are still far from understanding the whole process, specially from the latter, and therefore the discovery of potential treatment options. Consequently, any new study including new case descriptions and novel data is welcome. In my opinion, the data presented here has been very well discussed, and compared to what has been reported so far. The paper includes proper references and is generally well written, although some paragraphs would need to be clarified for a better understanding.

I think this paper would certainly be of interest for the typical readership of Journal of Clinical Medicine and I recommend it for publication after some changes:

  1. Protocols are well described although I suggest to include a summarizing table of all the antibodies used in the study (reference, dilution, purpose…). This is always helpful for future readers.
  2. Line 50. I think reference 3 does not include any suitable example in reference to animal/cell models. It should be removed.
  3. Unit annotation should be reviewed carefully and standardized throughout the whole text and supplementary data.

Example: line 150; 10 mM and line 158; 40mg/mL.  Also, lines 260 311, 316…

  1. Proper annotation should include a space between a numeral and a unit; i.e: 10 nM.
  2. Line 87: revise spaces and commas throughout the text and supplementary materials; i.e. MVID(38), Golgistacks/TGN.
  3. Line 325: Indicate in the title that the monoclonal antibody #596 works against CFTR so that is it clear to the reader.
  4. Lines 408 and 447: indicate that protocol I, II, III… are defined in supplementary materials and methods, at least the first time they are mentioned.
  5. Lines 456-457: As you show, organoids represent a very close model to what happens in the organ but, could it be that some features cannot be recapitulated? In that sense, have you observed alterations of the basolateral domain of enteroids such as ectopic villi or interdigitations as well? Please, mention that this feature cannot be recapitulated and include it here if you have a possible explanation for this.
  6. Lines 465-469: You have previously shown that dilated TGN was observed in more than half of the villus enterocytes, however you did not find this feature in enteroids (line 466). Is there any chance that this could represent an “artifact” of the technique employed? (standard procedures-protocol I for patient sample vs cryo-fixation protocol II for enteroids).

Could it be related to the different degrees of maturation that enteroids show, in comparison to the full differentiated villus enterocytes? This fact could explain the point number 8 too. I think you should include a brief explanation for this.

  1. Lines 458-459: you refer to “control samples” but you do not include any Figure showing these data. A picture of a control enteroid should be included at least in supplementary materials, so that these general features can be compared. Similarly, in lines 411-412, a “control sample” is mentioned but any picture has been included. A picture of a control enterocyte should be also added.
  2. Lines 471-485: this whole paragraph is very confusing and needs to be rewritten, especially after lines 478 and 479 the reading cannot continue fluently. If I understand correctly, the subapical region contains a wide variety of vesicles: 1) A great accumulation of abnormal subapical vesicles, found PAS-positive, Rab8-Rab11 positive. This is in accordance to previous observations. 2) Slim vesicles that, according to previous observation, represent unaltered RE. However, observations regarding PAS-positive vesicles and vesicle shapes are described in lines 418-421 (MVID patient sample) and 492-495 (enteroid?) which makes the reader going back and forward. Please, make all this information clear or separate it in different chapters so that it can be read fluently.
  3. Line 496: indicate Supplementary Movie 1 together with Figure2g-i.
  4. Line 554: Nomenclature of genes/proteins should be carefully reviewed as well as the use of uppercase/lowercase. i.e: rab11/Rab11a.
  5. Line 585: method I means protocol I? please, standardized.
  6. Supplementary Table I: correct spelling mistakes (DPPI4). M6PR in liver does not have an X, please include or remove. Indicate that BSEP and MRP2 were not studied in duodenum biopsies and the same for intestinal markers not studied in liver.
  7. Figure 4: yellow and orange colors are very similar and hard to distinguish. I recommend to change color patters as for example you did on your previous publication (Ref 9) or Fig. 4.
  8. Line 803: what do you mean by “the surface increases”? I understand that this is related to what has been indicated in lines 409-411 and 499-800 and you mean “absorptive surface”, please clarify in the text.

Author Response

Thank you very much for the detailed and quick evaluation. The hints for improvement  were very helpfull. Our point by point reply is attached, text changes are highlighted in the manuscript in yellow. 
